# Subset Selection by Pareto Optimization

**Chao Qian**       **Yang Yu**       **Zhi-Hua Zhou**
National Key Laboratory for Novel Software Technology, Nanjing University
Collaborative Innovation Center of Novel Software Technology and Industrialization
Nanjing 210023, China
{qianc,yuy,zhouzh}@lamda.nju.edu.cn

## Abstract

Selecting the optimal subset from a large set of variables is a fundamental problem in various learning tasks such as feature selection, sparse regression, dictionary learning, etc. In this paper, we propose the POSS approach which employs evolutionary Pareto optimization to find a small-sized subset with good performance. We prove that for sparse regression, POSS is able to achieve the best-so-far theoretically guaranteed approximation performance efficiently. Particularly, for the *Exponential Decay* subclass, POSS is proven to achieve an optimal solution. Empirical study verifies the theoretical results, and exhibits the superior performance of POSS to greedy and convex relaxation methods.

## 1 Introduction

Subset selection is to select a subset of size $k$ from a total set of $n$ variables for optimizing some criterion. This problem arises in many applications, e.g., feature selection, sparse learning and compressed sensing. The subset selection problem is, however, generally NP-hard [13, 4]. Previous employed techniques can be mainly categorized into two branches, greedy algorithms and convex relaxation methods. Greedy algorithms iteratively select or abandon one variable that makes the criterion currently optimized [9, 19], which are however limited due to its greedy behavior. Convex relaxation methods usually replace the set size constraint (i.e., the $\ell_0$-norm) with convex constraints, e.g., the $\ell_1$-norm constraint [18] and the elastic net penalty [29]; then find the optimal solutions to the relaxed problem, which however could be distant to the true optimum.

Pareto optimization solves a problem by reformulating it as a bi-objective optimization problem and employing a bi-objective evolutionary algorithm, which has significantly developed recently in theoretical foundation [22, 15] and applications [16]. This paper proposes the POSS (Pareto Optimization for Subset Selection) method, which treats subset selection as a bi-objective optimization problem that optimizes some given criterion and the subset size simultaneously. To investigate the performance of POSS, we study a representative example of subset selection, the sparse regression.

The subset selection problem in sparse regression is to best estimate a predictor variable by linear regression [12], where the quality of estimation is usually measured by the mean squared error, or equivalently, the squared multiple correlation $R^2$ [6, 11]. Gilbert et al. [9] studied the two-phased approach with orthogonal matching pursuit (OMP), and proved the multiplicative approximation guarantee $1 + \Theta(\mu k^2)$ for the mean squared error, when the coherence $\mu$ (i.e., the maximum correlation between any pair of observation variables) is $O(1/k)$. This approximation bound was later improved by [20, 19]. Under the same small coherence condition, Das and Kempe [2] analyzed the forward regression (FR) algorithm [12] and obtained an approximation guarantee $1 - \Theta(\mu k)$ for $R^2$. These results however will break down when $\mu \in w(1/k)$. By introducing the submodularity ratio $\gamma$, Das and Kempe [3] proved the approximation guarantee $1 - e^{-\gamma}$ on $R^2$ by the FR algorithm; this guarantee is considered to be the strongest since it can be applied with any coherence. Note that sparse regression is similar to the problem of sparse recovery [7, 25, 21, 17], but they are for

different purposes. Assuming that the predictor variable has a sparse representation, sparse recovery is to recover the exact coefficients of the truly sparse solution.

We theoretically prove that, for sparse regression, POSS using polynomial time achieves a multiplicative approximation guarantee $1 - e^{-\gamma}$ for squared multiple correlation $R^2$, the best-so-far guarantee obtained by the FR algorithm [3]. For the Exponential Decay subclass, which has clear applications in sensor networks [2], POSS can provably find an optimal solution, while FR cannot. The experimental results verify the theoretical results and exhibit the superior performance of POSS.

We start the rest of the paper by introducing the subset selection problem. We then present in three subsequent sections the POSS method, its theoretical analysis for sparse regression, and the empirical studies. The final section concludes this paper.

## 2 Subset Selection

The subset selection problem originally aims at selecting a few columns from a matrix, so that the matrix is most represented by the selected columns [1]. In this paper, we present the generalized subset selection problem that can be applied to arbitrary criterion evaluating the selection.

### 2.1 The General Problem

Given a set of observation variables $V = \{X_1, \ldots, X_n\}$, a criterion $f$ and a positive integer $k$, the subset selection problem is to select a subset $S \subseteq V$ such that $f$ is optimized with the constraint $|S| \leq k$, where $|\cdot|$ denotes the size of a set. For notational convenience, we will not distinguish between $S$ and its index set $I_S = \{i \mid X_i \in S\}$. Subset selection is formally stated as follows.

**Definition 1** (Subset Selection). *Given all variables $V = \{X_1, \ldots, X_n\}$, a criterion $f$ and a positive integer $k$, the subset selection problem is to find the solution of the optimization problem:*

$$\arg\min_{S \subseteq V} f(S) \quad s.t. \quad |S| \leq k. \tag{1}$$

The subset selection problem is NP-hard in general [13, 4], except for some extremely simple criteria. In this paper, we take sparse regression as the representative case.

### 2.2 Sparse Regression

Sparse regression [12] finds a sparse approximation solution to the regression problem, where the solution vector can only have a few non-zero elements.

**Definition 2** (Sparse Regression). *Given all observation variables $V = \{X_1, \ldots, X_n\}$, a predictor variable $Z$ and a positive integer $k$, define the* mean squared error *of a subset $S \subseteq V$ as*

$$MSE_{Z,S} = \min_{\boldsymbol{\alpha} \in \mathbb{R}^{|S|}} \mathbb{E}\left[\left(Z - \sum\nolimits_{i \in S} \alpha_i X_i\right)^2\right].$$

*Sparse regression is to find a set of at most $k$ variables minimizing the mean squared error, i.e.,*

$$\arg\min_{S \subseteq V} MSE_{Z,S} \quad s.t. \quad |S| \leq k.$$

For the ease of theoretical treatment, the *squared multiple correlation*

$$R^2_{Z,S} = (Var(Z) - MSE_{Z,S})/Var(Z)$$

is used to replace $MSE_{Z,S}$ [6, 11] so that the sparse regression is equivalently

$$\arg\max_{S \subseteq V} R^2_{Z,S} \quad s.t. \quad |S| \leq k. \tag{2}$$

Sparse regression is a representative example of subset selection [12]. Note that we will study Eq. (2) in this paper. Without loss of generality, we assume that all random variables are normalized to have expectation 0 and variance 1. Thus, $R^2_{Z,S}$ is simplified to be $1 - MSE_{Z,S}$.

For sparse regression, Das and Kempe [3] proved that the forward regression (FR) algorithm, presented in Algorithm 1, can produce a solution $S^{FR}$ with $|S^{FR}| = k$ and $R^2_{Z,SFR} \geq (1 - e^{-\gamma_{SFR,k}}) \cdot OPT$ (where $OPT$ denotes the optimal function value of Eq. (2)), which is the best currently known approximation guarantee. The FR algorithm is a greedy approach, which iteratively selects a variable with the largest $R^2$ improvement.

---

**Algorithm 1** Forward Regression

---

**Input**: all variables $V = \{X_1, \ldots, X_n\}$, a predictor variable $Z$ and an integer parameter $k \in [1, n]$
**Output**: a subset of $V$ with $k$ variables
**Process**:
1: Let $t = 0$ and $S_t = \emptyset$.
2: **repeat**
3:  Let $X^*$ be a variable maximizing $R^2_{Z, S_t \cup \{X\}}$, i.e., $X^* = \arg\max_{X \in V \setminus S_t} R^2_{Z, S_t \cup \{X\}}$.
4:  Let $S_{t+1} = S_t \cup \{X^*\}$, and $t = t + 1$.
5: **until** $t = k$
6: **return** $S_k$

---

## 3  The POSS Method

The subset selection in Eq. (1) can be separated into two objectives, one optimizes the criterion, i.e., $\min_{S \subseteq V} f(S)$, meanwhile the other keeps the size small, i.e., $\min_{S \subseteq V} \max\{|S| - k, 0\}$. Usually the two objectives are conflicting, that is, a subset with a better criterion value could have a larger size. The POSS method solves the two objectives simultaneously, which is described as follows.

Let us use the binary vector representation for subsets membership indication, i.e., $\boldsymbol{s} \in \{0, 1\}^n$ represents a subset $S$ of $V$ by assigning $s_i = 1$ if the $i$-th element of $V$ is in $S$ and $s_i = 0$ otherwise. We assign two properties for a solution $\boldsymbol{s}$: $o_1$ is the criterion value and $o_2$ is the sparsity,

$$\boldsymbol{s}.o_1 = \begin{cases} +\infty, & \boldsymbol{s} = \{0\}^n, \text{ or } |\boldsymbol{s}| \geq 2k \\ f(\boldsymbol{s}), & \text{otherwise} \end{cases}, \quad \boldsymbol{s}.o_2 = |\boldsymbol{s}|.$$

where the set of $o_1$ to $+\infty$ is to exclude trivial or overly bad solutions. We further introduce the isolation function $I : \{0, 1\}^n \to \mathbb{R}$ as in [22], which determines if two solutions are allowed to be compared: they are comparable only if they have the same isolation function value. The implementation of $I$ is left as a parameter of the method, while its effect will be clear in the analysis.

As will be introduced later, we need to compare solutions. For solutions $\boldsymbol{s}$ and $\boldsymbol{s}'$, we first judge if they have the same isolation function value. If not, we say that they are *incomparable*. If they have the same isolation function value, $\boldsymbol{s}$ is *worse* than $\boldsymbol{s}'$ if $\boldsymbol{s}'$ has a smaller or equal value on both the properties; $\boldsymbol{s}$ is *strictly worse* if $\boldsymbol{s}'$ has a strictly smaller value in one property, and meanwhile has a smaller or equal value in the other property. But if both $\boldsymbol{s}$ is not worse than $\boldsymbol{s}'$ and $\boldsymbol{s}'$ is not worse than $\boldsymbol{s}$, we still say that they are *incomparable*.

POSS is described in Algorithm 2. Starting from the solution representing an empty set and the archive $P$ containing only the empty set (line 1), POSS generates new solutions by randomly flipping bits of an archived solution (in the binary vector representation), as lines 4 and 5. Newly generated solutions are compared with the previously archived solutions (line 6). If the newly generated solution is not strictly worse than any previously archived solution, it will be archived. Before archiving the newly generated solution in line 8, the archive set $P$ is cleaned by removing solutions in $Q$, which are previously archived solutions but are worse than the newly generated solution.

The iteration of POSS repeats for $T$ times. Note that $T$ is a parameter, which could depend on the available resource of the user. We will analyze the relationship between the solution quality and $T$ in later sections, and will use the theoretically derived $T$ value in the experiments. After the iterations, we select the final solution from the archived solutions according to Eq. (1), i.e., select the solution with the smallest $f$ value while the constraint on the set size is kept (line 12).

## 4  POSS for Sparse Regression

In this section, we examine the theoretical performance of the POSS method for sparse regression. For sparse regression, the criterion $f$ is implemented as $f(\boldsymbol{s}) = -R^2_{Z, \boldsymbol{s}}$. Note that minimizing $-R^2_{Z, \boldsymbol{s}}$ is equivalent to the original objective that maximizes $R^2_{Z, \boldsymbol{s}}$ in Eq. (2).

We need some notations for the analysis. Let $Cov(\cdot, \cdot)$ be the covariance between two random variables, $C$ be the covariance matrix between all observation variables, i.e., $C_{i,j} = Cov(X_i, X_j)$,

---

**Algorithm 2** POSS

---

**Input**: all variables $V = \{X_1, \ldots, X_n\}$, a given criterion $f$ and an integer parameter $k \in [1, n]$
**Parameter**: the number of iterations $T$ and an isolation function $I : \{0, 1\}^n \to \mathbb{R}$
**Output**: a subset of $V$ with at most $k$ variables
**Process**:

1: Let $\boldsymbol{s} = \{0\}^n$ and $P = \{\boldsymbol{s}\}$.
2: Let $t = 0$.
3: **while** $t < T$ **do**
4:     Select $\boldsymbol{s}$ from $P$ uniformly at random.
5:     Generate $\boldsymbol{s}'$ from $\boldsymbol{s}$ by flipping each bit of $\boldsymbol{s}$ with probability $1/n$.
6:     **if** $\nexists \boldsymbol{z} \in P$ such that $I(\boldsymbol{z}) = I(\boldsymbol{s}')$ and $\big((\boldsymbol{z}.o_1 < \boldsymbol{s}'.o_1 \land \boldsymbol{z}.o_2 \leq \boldsymbol{s}'.o_2)$ or $(\boldsymbol{z}.o_1 \leq \boldsymbol{s}'.o_1 \land$
        $\boldsymbol{z}.o_2 < \boldsymbol{s}'.o_2)\big)$ **then**
7:         $Q = \{\boldsymbol{z} \in P \mid I(\boldsymbol{z}) = I(\boldsymbol{s}') \land \boldsymbol{s}'.o_1 \leq \boldsymbol{z}.o_1 \land \boldsymbol{s}'.o_2 \leq \boldsymbol{z}.o_2\}$.
8:         $P = (P \setminus Q) \cup \{\boldsymbol{s}'\}$.
9:     **end if**
10:    $t = t + 1$.
11: **end while**
12: **return** $\arg\min_{\boldsymbol{s} \in P, |\boldsymbol{s}| \leq k} f(\boldsymbol{s})$

---

and $\boldsymbol{b}$ be the covariance vector between $Z$ and observation variables, i.e., $b_i = Cov(Z, X_i)$. Let $C_S$ denote the submatrix of $C$ with row and column set $S$, and $\boldsymbol{b}_S$ denote the subvector of $\boldsymbol{b}$, containing elements $b_i$ with $i \in S$. Let $Res(Z, S) = Z - \sum_{i \in S} \alpha_i X_i$ denote the *residual* of $Z$ with respect to $S$, where $\boldsymbol{\alpha} \in \mathbb{R}^{|S|}$ is the least square solution to $MSE_{Z,S}$ [6]. The submodularity ratio presented in Definition 3 is a measure characterizing how close a set function $f$ is to submodularity. It is easy to see that $f$ is submodular iff $\gamma_{U,k}(f) \geq 1$ for any $U$ and $k$. For $f$ being the objective function $R^2$, we will use $\gamma_{U,k}$ shortly in the paper.

**Definition 3** (Submodularity Ratio [3]). *Let $f$ be a non-negative set function. The submodularity ratio of $f$ with respect to a set $U$ and a parameter $k \geq 1$ is*

$$\gamma_{U,k}(f) = \min_{L \subseteq U, S : |S| \leq k, S \cap L = \emptyset} \frac{\sum_{x \in S}(f(L \cup \{x\}) - f(L))}{f(L \cup S) - f(L)}.$$

### 4.1 On General Sparse Regression

Our first result is the theoretical approximation bound of POSS for sparse regression in Theorem 1. Let $OPT$ denote the optimal function value of Eq. (2). The expected running time of POSS is the average number of objective function (i.e., $R^2$) evaluations, the most time-consuming step, which is also the average number of iterations $T$ (denoted by $\mathbb{E}[T]$) since it only needs to perform one objective evaluation for the newly generated solution $\boldsymbol{s}'$ in each iteration.

**Theorem 1.** *For sparse regression, POSS with $\mathbb{E}[T] \leq 2ek^2n$ and $I(\cdot) = 0$ (i.e., a constant function) finds a set $S$ of variables with $|S| \leq k$ and $R^2_{Z,S} \geq (1 - e^{-\gamma_{\emptyset,k}}) \cdot OPT$.*

The proof relies on the property of $R^2$ in Lemma 1, that for any subset of variables, there always exists another variable, the inclusion of which can bring an improvement on $R^2$ proportional to the current distance to the optimum. Lemma 1 is extracted from the proof of Theorem 3.2 in [3].

**Lemma 1.** *For any $S \subseteq V$, there exists one variable $\hat{X} \in V - S$ such that*

$$R^2_{Z,S \cup \{\hat{X}\}} - R^2_{Z,S} \geq \frac{\gamma_{\emptyset,k}}{k}(OPT - R^2_{Z,S}).$$

*Proof.* Let $S^*_k$ be the optimal set of variables of Eq. (2), i.e., $R^2_{Z,S^*_k} = OPT$. Let $\bar{S} = S^*_k - S$ and $S' = \{Res(X, S) \mid X \in \bar{S}\}$. Using Lemmas 2.3 and 2.4 in [2], we can easily derive that $R^2_{Z,S \cup \bar{S}} = R^2_{Z,S} + R^2_{Z,S'}$. Because $R^2_{Z,S}$ increases with $S$ and $S^*_k \subseteq S \cup \bar{S}$, we have $R^2_{Z,S \cup \bar{S}} \geq R^2_{Z,S^*_k} = OPT$. Thus, $R^2_{Z,S'} \geq OPT - R^2_{Z,S}$. By Definition 3, $|S'| = |\bar{S}| \leq k$ and $R^2_{Z,\emptyset} = 0$, we get $\sum_{X' \in S'} R^2_{Z,X'} \geq \gamma_{\emptyset,k} R^2_{Z,S'} \geq \gamma_{\emptyset,k}(OPT - R^2_{Z,S})$. Let $\hat{X}' = \arg\max_{X' \in S'} R^2_{Z,X'}$. Then, $R^2_{Z,\hat{X}'} \geq \frac{\gamma_{\emptyset,k}}{|S'|}(OPT - R^2_{Z,S}) \geq \frac{\gamma_{\emptyset,k}}{k}(OPT - R^2_{Z,S})$. Let $\hat{X} \in \bar{S}$ correspond to $\hat{X}'$, i.e., $Res(\hat{X}, S) = \hat{X}'$. Thus, $R^2_{Z,S \cup \{\hat{X}\}} - R^2_{Z,S} = R^2_{Z,\hat{X}'} \geq \frac{\gamma_{\emptyset,k}}{k}(OPT - R^2_{Z,S})$. The lemma holds. $\qquad\square$

**Proof of Theorem 1.** Since the isolation function is a constant function, all solutions are allowed to be compared and we can ignore it. Let $J_{\max}$ denote the maximum value of $j \in [0, k]$ such that in the archive set $P$, there exists a solution $\boldsymbol{s}$ with $|\boldsymbol{s}| \leq j$ and $R^2_{Z,\boldsymbol{s}} \geq (1 - (1 - \frac{\gamma_{\emptyset,k}}{k})^j) \cdot OPT$. That is, $J_{\max} = \max\{j \in [0, k] \mid \exists \boldsymbol{s} \in P, |\boldsymbol{s}| \leq j \wedge R^2_{Z,\boldsymbol{s}} \geq (1 - (1 - \frac{\gamma_{\emptyset,k}}{k})^j) \cdot OPT\}$. We then analyze the expected iterations until $J_{\max} = k$, which implies that there exists one solution $\boldsymbol{s}$ in $P$ satisfying that $|\boldsymbol{s}| \leq k$ and $R^2_{Z,\boldsymbol{s}} \geq (1 - (1 - \frac{\gamma_{\emptyset,k}}{k})^k) \cdot OPT \geq (1 - e^{-\gamma_{\emptyset,k}}) \cdot OPT$.

The initial value of $J_{\max}$ is 0, since POSS starts from $\{0\}^n$. Assume that currently $J_{\max} = i < k$. Let $\boldsymbol{s}$ be a corresponding solution with the value $i$, i.e., $|\boldsymbol{s}| \leq i$ and $R^2_{Z,\boldsymbol{s}} \geq (1 - (1 - \frac{\gamma_{\emptyset,k}}{k})^i) \cdot OPT$. It is easy to see that $J_{\max}$ cannot decrease because cleaning $\boldsymbol{s}$ from $P$ (lines 7 and 8 of Algorithm 2) implies that $\boldsymbol{s}$ is "worse" than a newly generated solution $\boldsymbol{s}'$, which must have a smaller size and a larger $R^2$ value. By Lemma 1, we know that flipping one specific 0 bit of $\boldsymbol{s}$ (i.e., adding a specific variable into $S$) can generate a new solution $\boldsymbol{s}'$, which satisfies that $R^2_{Z,\boldsymbol{s}'} - R^2_{Z,\boldsymbol{s}} \geq \frac{\gamma_{\emptyset,k}}{k}(OPT - R^2_{Z,\boldsymbol{s}})$. Then, we have

$$R^2_{Z,\boldsymbol{s}'} \geq (1 - \frac{\gamma_{\emptyset,k}}{k})R^2_{Z,\boldsymbol{s}} + \frac{\gamma_{\emptyset,k}}{k} \cdot OPT \geq (1 - (1 - \frac{\gamma_{\emptyset,k}}{k})^{i+1}) \cdot OPT.$$

Since $|\boldsymbol{s}'| = |\boldsymbol{s}| + 1 \leq i + 1$, $\boldsymbol{s}'$ will be included into $P$; otherwise, from line 6 of Algorithm 2, $\boldsymbol{s}'$ must be "strictly worse" than one solution in $P$, and this implies that $J_{\max}$ has already been larger than $i$, which contradicts with the assumption $J_{\max} = i$. After including $\boldsymbol{s}'$, $J_{\max} \geq i+1$. Let $P_{\max}$ denote the largest size of $P$. Thus, $J_{\max}$ can increase by at least 1 in one iteration with probability at least $\frac{1}{P_{\max}} \cdot \frac{1}{n}(1 - \frac{1}{n})^{n-1} \geq \frac{1}{enP_{\max}}$, where $\frac{1}{P_{\max}}$ is a lower bound on the probability of selecting $\boldsymbol{s}$ in line 4 of Algorithm 2 and $\frac{1}{n}(1 - \frac{1}{n})^{n-1}$ is the probability of flipping a specific bit of $\boldsymbol{s}$ and keeping other bits unchanged in line 5. Then, it needs at most $enP_{\max}$ expected iterations to increase $J_{\max}$. Thus, after $k \cdot enP_{\max}$ expected iterations, $J_{\max}$ must have reached $k$.

By the procedure of POSS, we know that the solutions maintained in $P$ must be incomparable. Thus, each value of one property can correspond to at most one solution in $P$. Because the solutions with $|\boldsymbol{s}| \geq 2k$ have $+\infty$ value on the first property, they must be excluded from $P$. Thus, $|\boldsymbol{s}| \in \{0, 1, \ldots, 2k - 1\}$, which implies that $P_{\max} \leq 2k$. Hence, the expected number of iterations $\mathbb{E}[T]$ for finding the desired solution is at most $2ek^2n$. $\qquad\square$

Comparing with the approximation guarantee of FR, $(1 - e^{-\gamma_{SFR,k}}) \cdot OPT$ [3], it is easy to see that $\gamma_{\emptyset,k} \geq \gamma_{SFR,k}$ from Definition 3. Thus, POSS with the simplest configuration of the isolation function can do at least as well as FR on any sparse regression problem, and achieves the best previous approximation guarantee. We next investigate if POSS can be strictly better than FR.

## 4.2 On The Exponential Decay Subclass

Our second result is on a subclass of sparse regression, called Exponential Decay as in Definition 4. In this subclass, the observation variables can be ordered in a line such that their covariances are decreasing exponentially with the distance.

**Definition 4** (Exponential Decay [2]). *The variables $X_i$ are associated with points $y_1 \leq y_2 \leq \ldots \leq y_n$, and $C_{i,j} = a^{|y_i - y_j|}$ for some constant $a \in (0, 1)$.*

Since we have shown that POSS with a constant isolation function is generally good, we prove below that POSS with a proper isolation function can be even better: it is strictly better than FR on the Exponential Decay subclass, as POSS finds an optimal solution (i.e., Theorem 2) while FR cannot (i.e., Proposition 1). The isolation function $I(\boldsymbol{s} \in \{0,1\}^n) = \min\{i \mid s_i = 1\}$ implies that two solutions are comparable only if they have the same minimum index for bit 1.

**Theorem 2.** *For the Exponential Decay subclass of sparse regression, POSS with $\mathbb{E}[T] \in O(k^2(n - k)n \log n)$ and $I(\boldsymbol{s} \in \{0,1\}^n) = \min\{i \mid s_i = 1\}$ finds an optimal solution.*

The proof of Theorem 2 utilizes the dynamic programming property of the problem, as in Lemma 2.

**Lemma 2.** *[2] Let $R^2(v, j)$ denote the maximum $R^2_{Z,S}$ value by choosing $v$ variables, including necessarily $X_j$, from $X_j, \ldots, X_n$. That is, $R^2(v, j) = \max\{R^2_{Z,S} \mid S \subseteq \{X_j, \ldots, X_n\}, X_j \in S, |S| = v\}$. Then, the following recursive relation holds:*

$$R^2(v + 1, j) = \max_{j+1 \leq i \leq n} \left( R^2(v, i) + b_j^2 + (b_j - b_i)^2 \frac{a^{2|y_i - y_j|}}{1 - a^{2|y_i - y_j|}} - 2b_j b_i \frac{a^{|y_i - y_j|}}{1 + a^{|y_i - y_j|}} \right),$$

*where the term in $()$ is the $R^2$ value by adding $X_j$ into the variable subset corresponding to $R^2(v, i)$.*

**Proof of Theorem 2.** We divide the optimization process into $k+1$ phases, where the $i$-th ($1 \leq i \leq k$) phase starts after the $(i-1)$-th phase has finished. We define that the $i$-th phase finishes when for each solution corresponding to $R^2(i,j)$ ($1 \leq j \leq n-i+1$), there exists one solution in the archive $P$ which is "better" than it. Here, a solution $s$ is "better" than $s'$ is equivalent to that $s'$ is "worse" than $s$. Let $\xi_i$ denote the iterations since phase $i-1$ has finished, until phase $i$ is completed.

Starting from the solution $\{0\}^n$, the 0-th phase has finished. Then, we consider $\xi_i$ ($i \geq 1$). In this phase, from Lemma 2, we know that a solution "better" than a corresponding solution of $R^2(i,j)$ can be generated by selecting a specific one from the solutions "better" than $R^2(i-1,j+1), \dots, R^2(i-1,n)$ and flipping its $j$-th bit, which happens with probability at least $\frac{1}{P_{\max}} \cdot \frac{1}{n}(1-\frac{1}{n})^{n-1} \geq \frac{1}{enP_{\max}}$. Thus, if we have found $L$ desired solutions in the $i$-th phase, the probability of finding a new desired solution in the next iteration is at least $(n-i+1-L) \cdot \frac{1}{enP_{\max}}$, where $n-i+1$ is the total number of desired solutions to find in the $i$-th phase. Then, $\mathbb{E}[\xi_i] \leq \sum_{L=0}^{n-i} \frac{enP_{\max}}{n-i+1-L} \in O(n\log nP_{\max})$. Therefore, the expected number of iterations $\mathbb{E}[T]$ is $O(kn\log nP_{\max})$ until the $k$-th phase finishes, which implies that an optimal solution corresponding to $\max_{1 \leq j \leq n} R^2(k,j)$ has been found. Note that $P_{\max} \leq 2k(n-k)$, because the incomparable property of the maintained solutions by POSS ensures that there exists at most one solution in $P$ for each possible combination of $|s| \in \{0,1,\dots,2k-1\}$ and $I(s) \in \{0,1,\dots,n\}$. Thus, $\mathbb{E}[T]$ for finding an optimal solution is $O(k^2(n-k)n\log n)$. $\quad\square$

Then, we analyze FR (i.e., Algorithm 1) for this special class. We show below that FR can be blocked from finding an optimal solution by giving a simple example.

**Example 1.** $X_1 = Y_1$, $X_i = r_i X_{i-1} + Y_i$, where $r_i \in (0,1)$, and $Y_i$ are independent random variables with expectation 0 such that each $X_i$ has variance 1.

For $i < j$, $Cov(X_i, X_j) = \prod_{k=i+1}^{j} r_k$. Then, it is easy to verify that Example 1 belongs to the Exponential Decay class by letting $y_1 = 0$ and $y_i = \sum_{k=2}^{i} \log_a r_k$ for $i \geq 2$.

**Proposition 1.** *For Example 1 with $n = 3$, $r_2 = 0.03$, $r_3 = 0.5$, $Cov(Y_1, Z) = Cov(Y_2, Z) = \delta$ and $Cov(Y_3, Z) = 0.505\delta$, FR cannot find the optimal solution for $k = 2$.*

*Proof.* The covariances between $X_i$ and $Z$ are $b_1 = \delta$, $b_2 = 0.03b_1 + \delta = 1.03\delta$ and $b_3 = 0.5b_2 + 0.505\delta = 1.02\delta$. Since $X_i$ and $Z$ have expectation 0 and variance 1, $R^2_{Z,S}$ can be simply represented as $b_S^T C_S^{-1} b_S$ [11]. We then calculate the $R^2$ value as follows: $R^2_{Z,X_1} = \delta^2$, $R^2_{Z,X_2} = 1.0609\delta^2$, $R^2_{Z,X_3} = 1.0404\delta^2$; $R^2_{Z,\{X_1,X_2\}} = 2.0009\delta^2$, $R^2_{Z,\{X_1,X_3\}} = 2.0103\delta^2$, $R^2_{Z,\{X_2,X_3\}} = 1.4009\delta^2$. The optimal solution for $k = 2$ is $\{X_1, X_3\}$. FR first selects $X_2$ since $R^2_{Z,X_2}$ is the largest, then selects $X_1$ since $R^2_{Z,\{X_2,X_1\}} > R^2_{Z,\{X_2,X_3\}}$; thus produces a local optimal solution $\{X_1, X_2\}$. $\quad\square$

It is also easy to verify that other two previous methods OMP [19] and FoBa [26] cannot find the optimal solution for this example, due to their greedy nature.

## 5 Empirical Study

We conducted experiments on 12 data sets[1] in Table 1 to compare POSS with the following methods:
- **FR** [12] iteratively adds one variable with the largest improvement on $R^2$.
- **OMP** [19] iteratively adds one variable that mostly correlates with the predictor variable residual.
- **FoBa** [26] is based on OMP but deletes one variable adaptively when beneficial. Set parameter $\nu = 0.5$, the solution path length is five times as long as the maximum sparsity level (i.e., $5 \times k$), and the last active set containing $k$ variables is used as the final selection [26].
- **RFE** [10] iteratively deletes one variable with the smallest weight by linear regression.
- **Lasso** [18], **SCAD** [8] and **MCP** [24] replaces the $\ell_0$ norm constraint with the $\ell_1$ norm penalty, the smoothly clipped absolute deviation penalty and the mimimax concave penalty, respectively. For implementing these methods, we use the SparseReg toolbox developed in [28, 27].

For POSS, we use $I(\cdot) = 0$ since it is generally good, and the number of iterations $T$ is set to be $\lfloor 2ek^2 n \rfloor$ as suggested by Theorem 1. To evaluate how far these methods are from the optimum, we also compute the optimal subset by exhaustive enumeration, denoted as **OPT**.

Table 1: The data sets.

| data set | #inst | #feat | data set | #inst | #feat | data set | #inst | #feat |
|---|---|---|---|---|---|---|---|---|
| *housing* | 506 | 13 | *sonar* | 208 | 60 | *clean1* | 476 | 166 |
| *eunite2001* | 367 | 16 | *triazines* | 186 | 60 | *w5a* | 9888 | 300 |
| *svmguide3* | 1284 | 21 | *coil2000* | 9000 | 86 | *gisette* | 7000 | 5000 |
| *ionosphere* | 351 | 34 | *mushrooms* | 8124 | 112 | *farm-ads* | 4143 | 54877 |

Table 2: The training $R^2$ value (mean±std.) of the compared methods on 12 data sets for $k = 8$. In each data set, '●/○' denote respectively that POSS is significantly better/worse than the corresponding method by the $t$-*test* [5] with confidence level $0.05$. '-' means that no results were obtained after running several days.

| Data set | OPT | POSS | FR | FoBa | OMP | RFE | MCP |
|---|---|---|---|---|---|---|---|
| housing | .7437±.0297 | .7437±.0297 | .7429±.0300● | .7423±.0301● | .7415±.0300● | .7388±.0304● | .7354±.0297● |
| eunite2001 | .8484±.0132 | .8482±.0132 | .8348±.0143● | .8442±.0144● | .8349±.0150● | .8424±.0153● | .8320±.0150● |
| svmguide3 | .2705±.0255 | .2701±.0257 | .2615±.0260● | .2601±.0279● | .2557±.0270● | .2136±.0325● | .2397±.0237● |
| ionosphere | .5995±.0326 | .5990±.0329 | .5920±.0352● | .5929±.0346● | .5921±.0353● | .5832±.0415● | .5740±.0348● |
| sonar | – | .5365±.0410 | .5171±.0440● | .5138±.0432● | .5112±.0425● | .4321±.0636● | .4496±.0482● |
| triazines | – | .4301±.0603 | .4150±.0592● | .4107±.0600● | .4073±.0591● | .3615±.0712● | .3793±.0584● |
| coil2000 | – | .0627±.0076 | .0624±.0076● | .0619±.0075● | .0619±.0075● | .0363±.0141● | .0570±.0075● |
| mushrooms | – | .9912±.0020 | .9909±.0021● | .9909±.0022● | .9909±.0022● | .6813±.1294● | .8652±.0474● |
| clean1 | – | .4368±.0300 | .4169±.0299● | .4145±.0309● | .4132±.0315● | .1596±.0562● | .3563±.0364● |
| w5a | – | .3376±.0267 | .3319±.0247● | .3341±.0258● | .3313±.0246● | .3342±.0276● | .2694±.0385● |
| gisette | – | .7265±.0098 | .7001±.0116● | .6747±.0145● | .6731±.0134● | .5360±.0318● | .5709±.0123● |
| farm-ads | – | .4217±.0100 | .4196±.0101● | .4170±.0113● | .4170±.0113● | – | .3771±.0110● |
| POSS: win/tie/loss | – | | 12/0/0 | 12/0/0 | 12/0/0 | 11/0/0 | 12/0/0 |

To assess each method on each data set, we repeat the following process 100 times. The data set is randomly and evenly split into a training set and a test set. Sparse regression is built on the training set, and evaluated on the test set. We report the average training and test $R^2$ values.

## 5.1 On Optimization Performance

Table 2 lists the training $R^2$ for $k = 8$, which reveals the optimization quality of the methods. Note that the results of Lasso, SCAD and MCP are very close, and we only report that of MCP due to the page limit. By the $t$-*test* [5] with significance level $0.05$, POSS is shown significantly better than all the compared methods on all data sets.

We plot the performance curves on two data sets for $k \leq 8$ in Figure 1. For *sonar*, OPT is calculated only for $k \leq 5$. We can observe that POSS tightly follows OPT, and has a clear advantage over the rest methods. FR, FoBa and OMP have close performances, while are much better than MCP, SCAD and Lasso. The bad performance of Lasso is consistent with the previous results in [3, 26]. We notice that, although the $\ell_1$ norm constraint is a tight convex relaxation of the $\ell_0$ norm constraint and can have good results in sparse recovery tasks, the performance of Lasso is not as good as POSS and greedy methods on most data sets. This is due to that, unlike assumed in sparse recovery tasks, there may not exist a sparse structure in the data sets. In this case, $\ell_1$ norm constraint can be a bad approximation of $\ell_0$ norm constraint. Meanwhile, $\ell_1$ norm constraint also shifts the optimization problem, making it hard to well optimize the original $R^2$ criterion.

Considering the running time (in the number of objective function evaluations), OPT does exhaustive search, thus needs $\binom{n}{k} \geq \frac{n^k}{k^k}$ time, which could be unacceptable for a slightly large data set. FR, FoBa and OMP are greedy-like approaches, thus are efficient and their running time are all in the order of $kn$. POSS finds the solutions closest to those of OPT, taking $2ek^2n$ time. Although POSS is slower by a factor of $k$, the difference would be small when $k$ is a small constant.

Since the $2ek^2n$ time is a theoretical upper bound for POSS being as good as FR, we empirically examine how tight this bound is. By selecting FR as the baseline, we plot the curve of the $R^2$ value over the running time for POSS on the two largest data sets *gisette* and *farm-ads*, as shown in Figure 2. We do not split the training and test set, and the curve for POSS is the average of 30 independent runs. The $x$-axis is in $kn$, the running time of FR. We can observe that POSS takes about only 14% and 23% of the theoretical time to achieve a better performance, respectively on the two data sets. This implies that POSS can be more efficient in practice than in theoretical analysis.

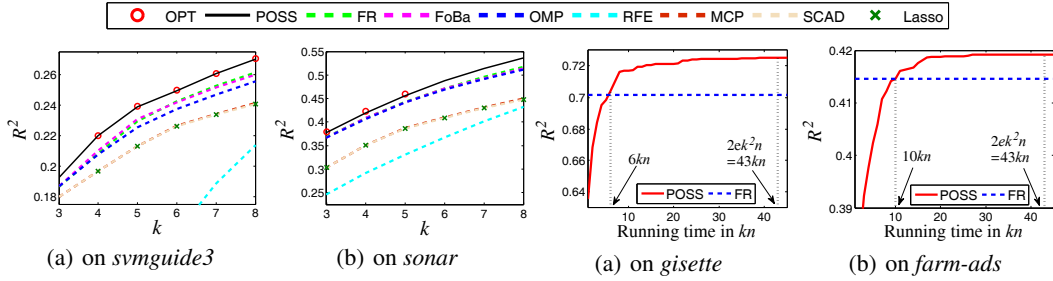

| (a) on *svmguide3* | (b) on *sonar* | (a) on *gisette* | (b) on *farm-ads* |

Figure 1: Training $R^2$ (the larger the better). Figure 2: Performance v.s. running time of POSS.

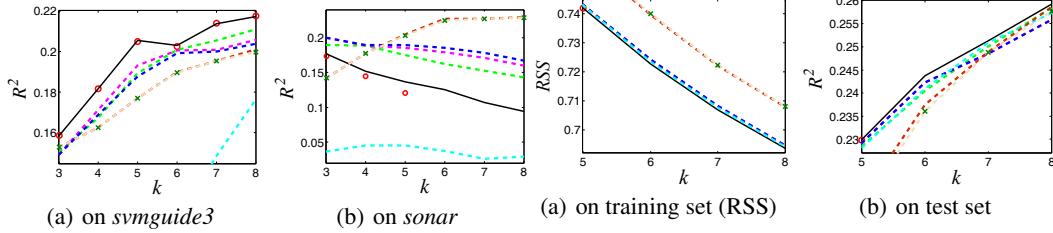

| (a) on *svmguide3* | (b) on *sonar* | (a) on training set (RSS) | (b) on test set |

Figure 3: Test $R^2$ (the larger the better). Figure 4: Sparse regression with $\ell_2$ regularization on *sonar*. $RSS$: the smaller the better.

## 5.2 On Generalization Performance

When testing sparse regression on the test data, it has been known that the sparsity alone may be not a good complexity measure [26], since it only restricts the number of variables, but the range of the variables is unrestricted. Thus better optimization does not always lead to better generalization performance. We also observe this in Figure 3. On *svmguide3*, test $R^2$ is consistent with training $R^2$ in Figure 1(a), however on *sonar*, better training $R^2$ (as in Figure 1(b)) leads to worse test $R^2$ (as in Figure 3(b)), which may be due to the small number of instances making it prone to overfitting.

As suggested in [26], other regularization terms may be necessary. We add the $\ell_2$ norm regularization into the objective function, i.e.,

$$RSS_{Z,S} = \min_{\boldsymbol{\alpha} \in \mathbb{R}^{|S|}} \mathbb{E}\left[ (Z - \sum_{i \in S} \alpha_i X_i)^2 \right] + \lambda |\boldsymbol{\alpha}|_2^2 .$$

The optimization is now $\arg\min_{S \subseteq V} RSS_{Z,S}$ *s.t.* $|S| \leq k$. We then test all the compared methods to solve this optimization problem with $\lambda = 0.9615$ on *sonar*. As plotted in Figure 4, we can observe that POSS still does the best optimization on the training RSS, and by introducing the $\ell_2$ norm, it leads to the best generalization performance in $R^2$.

## 6 Conclusion

In this paper, we study the problem of subset selection, which has many applications ranging from machine learning to signal processing. The general goal is to select a subset of size $k$ from a large set of variables such that a given criterion is optimized. We propose the POSS approach that solves the two objectives of the subset selection problem simultaneously, i.e., optimizing the criterion and reducing the subset size.

On sparse regression, a representative of subset selection, we theoretically prove that a simple POSS (i.e., using a constant isolation function) can generally achieve the best previous approximation guarantee, using time $2ek^2n$. Moreover, we prove that, with a proper isolation function, it finds an optimal solution for an important subclass *Exponential Decay* using time $O(k^2(n-k)n\log n)$, while other greedy-like methods may not find an optimal solution. We verify the superior performance of POSS by experiments, which also show that POSS can be more efficient than its theoretical time.

We will further study Pareto optimization from the aspects of using potential heuristic operators [14] and utilizing infeasible solutions [23]; and try to apply it to more machine learning tasks.

**Acknowledgements** We want to thank Lijun Zhang and Jianxin Wu for their helpful comments. This research was supported by 973 Program (2014CB340501) and NSFC (61333014, 61375061).

## Footnotes

[1] The data sets are from `http://archive.ics.uci.edu/ml/` and `http://www.csie.ntu.edu.tw/~cjlin/libsvmtools/datasets/`. Some binary classification data are used for regression. All variables are normalized to have mean 0 and variance 1.

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
