[Reviews · NeurIPS 2015]

Submitted by Assigned_Reviewer_1

In this paper, the authors apply Pareto optimization to the subset selection problem, specifically, sparse regression. A special case of sparse regression - exponential decay is discussed, where an optimal solution can be provably found by the proposed method.

Overall the presentation of the paper is clear. The method of subset selection looks not new, being close to the approach of [12], which also solves a subset selection problem. This paper focuses on sparse regression, and the main contribution is the theoretical analysis of POSS for sparse regression. The discussion of the special case of exponential decay is also novel and interesting. The theoretical analysis is sound. In the experiments, the authors may consider comparing to more sophisticated sparse regression methods such as MCP, SCAD and reweighted L1 norm.
Summary: In this paper, the authors apply Pareto optimization to the subset selection problem, specifically, sparse regression. The main contribution is the theoretical analysis of POSS for sparse regression, with a special case of exponential decay, where an optimal solution can be provably found by POSS. The theoretical analysis is sound.

Submitted by Assigned_Reviewer_2

The paper is concerned with a new algorithm (named POSS) for small-sized subset selection based on Pareto optimization. The proposed algorithm is extensively analyzed, compared and tested. A detailed comparison is presented between POSS and the FR algorithm (Forward Regression) for which a theoretical approximation bound is known. The authors also give such a theoretical approximation bound for their algorithm and conclude that POSS does at least as well as the FR algorithm on any sparse regression problem. A second result is given on the exponential decay subclass (a subclass of sparse regression: the observation variables can be ordered in a line such that their covariances are decreasing exponentially with the distance): On this subclass POSS has strictly better guarantees than FR. The authors also show that POSS finds (again on the exponential decay subclass) an optimal solution. This guarantee cannot be given for the FR algorithm as demonstrated be the authors. This concludes the theoretical part which is followed by an empirical study: 4 different algorithms (POSS, FR, OMP, FoBa) and the (exact) optimal solution (OPT), where it could be calculated in acceptable time, are compared on 12 different data sets. POSS outperforms all 4 algorithms when calculating the squared multiple correlation (R2) which is equivalent to the mean squared error. Although the authors claim a significantly better performance in all 4 cases, the reviewer is still not fully convinced about the attribute 'significantly'. The empirical study is extensive: calculating R2 on test and training data, performance vs running-time analysis and discussion of a possible pitfall (sparsity alone may not be a good sparsity measure and thus better optimization does not always lead to better generalization performance) and a solution to that pitfall (introduce l2 norm regularization into the objective function).

In general, the paper is written in a clear and transparent way, and it is easy to follow the main arguments.

Minor Comment: Line 126 - shouldn't it spell 'subset' instead of 'subsets'?
Summary: This topic of this paper is certainly relevant for the Machine Learning community. To my understanding, the results obtained are original and novel. An interesting theory paper.

Submitted by Assigned_Reviewer_3

The paper describes an interesting algorithm for supervised feature selection. The idea is to keep a pool of candidate subsets, and randomly perturb them. The perturbation creates a new subset by adding or

subtracting one variable (on average). Imposing an order on the subsets enables pruning and keeping the pool small. After a pre-determined number

of such perturbations the best candidate in the pool is produced as the result. The authors provide an analysis of worst case performance guarantee which is better than similar performance guarantees of some greedy algorithms.

To the best of my knowledge the algorithm is new and I believe it is very interesting. I would like to point out the following issues that the authors may want to consider.

1. The authors use "optimality" and "best" in a confusing way. Here is an example. The claim in the abstract that POSS has "the best theoretically guaranteed approximation performance" does not appear to be correct. It is only true when compared to some greedy algorithms. For example, backward elimination can be shown to be optimal when the optimal error is sufficiently small. Similar claims are made in the conclusions.

2. For the exponential decay case the optimality claim appears to be correct. But it is important to point out that dynamic programming can produce

the optimal solution without going through POSS, as shown in [3]. In fact, the algorithm of [3] appears to be significantly faster. This should, at the very least, be discussed.

3. The experimental evaluation compares POSS only with greedy algorithms. Since POSS is not greedy, this doesn't seem to be "fair". How does it compare, for example, with Guyon's RFE? In addition, since FR is known to be better than OMP, the comparison to both is of questionable value.
Summary: The paper describes an interesting algorithm for supervised feature selection. The algorithm is NOT greedy.

Analysis shows an improved worst case

performance guarantee when compared to some commonly used greedy algorithms. However, there is no comparison with other non-greedy algorithms. The paper appears to be heavily influenced by previous work of Das and Kempe.

Submitted by Assigned_Reviewer_4

This paper proposed a novel optimization algorithm for sparse regression using the framework of the subset selection. The paper theoretically showed its better lower bound of the maximization than the past methods. The experimental results demonstrate it superior performance in terms of the optimization errors.

The proposed algorithm and its associated theoretical analyses on its optimization bound have clear originality. The quality of the theoretical analysis and the experimental analysis on the optimization accuracy seems good. The paper is well and clearly written. The algorithm presented in this paper may have a considerable impact to the research field of the sparse regression.

However, the paper did not strictly argue its theoretical computational complexity and did not compare with these of the past representative methods. In addition, it did not provide the experimental comparison of the computation times. Since the trade off between the efficiency and the accuracy of the optimization algorithms is an important issue. These should be definitely added.

Minor Comment Line 6 of Algorithm 2 (z:o1

s0:o1 and z:o2

s0:o2) or ( ... ) seems to be duplicated.
Summary: This paper proposed a novel subset selection algorithm named POSS having a better optimization bound for the sparse regression. Though the proposed method and its characterization show clear significance, the title of the paper and the argument in the introduction seem not to correctly reflect the content of the

paper.

Submitted by Assigned_Reviewer_5

The paper proposed a new algorithm for subset selection based on Pareto optimization. The idea presented in the paper is interesting and intuitive. The results presented in the paper looks promising. Below are my comments to the paper:

1. When k is large for instance 200, k^2 * n * cost(inner_loop) could be a significant cost for performing subset selection. The authors need to provide a guideline for how to handle this case in the paper.

2. For performance evaluation, it is also interesting to see the performance on the test data, for example test R^2 obtained from cross validation.
Summary: The paper proposed a new algorithm for subset selection based on Pareto optimization. The idea presented in the paper is interesting and intuitive.

Author Feedback
Author rebuttal: To Reviewer1:

Comment: "The method of subset selection looks not new, being close to the approach of [12], which also solves a subset selection problem."
Comment: "In the experiments, the authors may consider comparing to more sophisticated sparse regression methods such as MCP, SCAD and reweighted L1 norm."

Response:
1. Our method is different from the approach of [12] in that we do not involve a local search subroutine as that in [12], and our approach has a much faster running time complexity. This will be discussed in the paper.
2. We have tested MCP and SCAD on the first ten data sets (the remaining data sets cost too much time for these methods), using the SparseReg toolbox developed by Hua Zhou. For MCP, the mean+std. of the training R^2 values are: 0.7354+0.0297, 0.8320+0.0150, 0.2397+0.0237, 0.5740+0.0348, 0.4496+0.0482, 0.3793+0.0584, 0.0570+0.0075, 0.8652+0.0474, 0.3563+0.0364, 0.2694+0.0385; and for SCAD, the results are very close. Comparing with the column of "POSS" in Table 2, POSS is consistently better than them. The complete results will be included in the paper.
Thank you for your comments.

To Reviewer2:

Comment: "1. The authors use "optimality" and "best" in a confusing way ... It is only true when compared to some greedy algorithms ..."
Comment: "2. For the exponential decay case the optimality claim appears to be correct. But it is important to point out that dynamic programming can produce the optimal solution ... faster. This should, at the very least, be discussed."
Comment: "3. The experimental evaluation compares POSS only with greedy algorithms. Since POSS is not greedy, this doesn't seem to be "fair". How does it compare, for example, with Guyon's RFE? In addition, since FR is known to be better than OMP, the comparison to both is of questionable value."

Response:
We have tested RFE on the first ten data sets (the remaining data sets cost too much time). The mean+std. of the training R^2 values are: 0.7388+0.0304, 0.8424+0.0153, 0.2136+0.0325, 0.5832+0.0415, 0.4321+0.0636, 0.3615+0.0712, 0.0363+0.0141, 0.6813+0.1294, 0.1596+0.0562, 0.3342+0.0276. Comparing with the column of "POSS" in Table 2, POSS is consistently better than RFE. The complete results will be included in the revision. And the language will be carefully revised.
Thank you very much.

To Reviewer3:

Comment: "1. When k is large for instance 200, k^2 * n * cost (inner_loop) could be a significant cost for performing subset selection. The authors need to provide a guideline for how to handle this case in the paper."
Comment: "2. For performance evaluation, it is also interesting to see the performance on the test data, for example test R^2 obtained from cross validation."

Response:
1. POSS is slower than greedy methods by a factor of k, but has a better optimization performance. When k is large, we could balance the efficiency and the efficacy by first running POSS for some features, e.g., k/2, and then running the greedy algorithm for the rest k/2 features. This will be discussed.
2. The test R^2 on data sets svmguide3 and sonar is shown in Figures 3 and 4. More will be included in the paper.
Thank you very much.

To Reviewer4:

Comment: "... it did not provide the experimental comparison of the computation times. Since the trade off between the efficiency and the accuracy of the optimization algorithms is an important issue. These should be definitely added."

Response:
The compared methods take as much time as they can, and their final results are reported, meanwhile, POSS can adjust the running time. Figure 2 shows optimization performance v.s. running time of POSS. More empirical computation time will be included in the paper.
Thank you very much.

To Reviewer6:

Comment: "... but I wonder whether similar methods were already proposed in non-machine learning literature such as in operations research."

Response: To the best of our knowledge, there is no previous work (except [12]) that uses Pareto optimization with a theoretical guarantee.
Thank you very much.

To Reviewer7:

We will revise the typos carefully. Thank you very much.